# Best Practices for Training in Sustainable Greenhouse Horticulture

**Ralph De Witte** [1], **Dirk Janssen** [1,*], **Samir Sayadi Gmada** [2] **and Carmen García-García** [1]

1   Instituto Andaluz de Investigación y Formación Agraria (IFAPA), Centro La Mojonera,
    04745 La Mojonera, Spain
2   Instituto Andaluz de Investigación y Formación Agraria (IFAPA), Centro Camino de Purchil,
    18004 Granada, Spain
*   Correspondence: dirk.janssen@juntadeandalucia.es; Tel.: +34-67-159-3012

**Abstract:** Consumer demands and current legislation require intensive greenhouse horticulture to be sustainable. This poses the challenge of how to teach the concept of sustainable horticulture to all professionals involved in farming. The province of Almeria, in the south-east of Spain, is one of the major horticulture greenhouse areas in Europe, and an expert panel of relevant stakeholders was invited to look into the best pedagogical practices and methods to transfer technology and knowledge, with the goal of improving the sustainability of greenhouse horticulture. A combination of an online questionnaire, a Delphi method, and desk research was, therefore, used as the strategy to collect the data and implement the research design during 2021. On-farm/business demonstrations, virtual education, and classroom education were common pedagogical methods used. On-farm/business demonstrations, participatory education, and co-learning were identified as the best pedagogical methods to use in sustainable agriculture/horticulture training. The expert panel also concluded that participatory education and co-learning should be further explored whereas virtual and classroom education should play a less dominant role in the training activities. This knowledge can help training organizations and designers to avoid common mistakes, tailor their training activities, and be mindful of common barriers and (mis)conceptions.

**Keywords:** farmer training; Delphi method; co-learning; horticultural greenhouses; sustainability; COVID-19

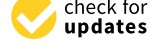



## 1. Introduction

Global demand for agricultural products is expected to double in the next decades, putting tremendous pressure on agriculture to produce more [1]. At the same time, there is an increasing concern about the effect that intensive agriculture has on biodiversity and climate change, resulting in a society-wide demand for the sustainable production of agriculture crops. These two phenomena are often seen as conflicting, yet there is another perspective that combines the two: sustainable agriculture—a method intended to protect the environment and improve the quality of a farmer's life while maintaining, or even increasing, the production level [2]. Coinciding with this new perspective comes the question and challenge of how to teach the concept of sustainable agriculture. Problems and barriers for sustainable agriculture identified before concern education and information [3–11], the management of information [3–13], and a lack of on-farm trials and demonstrations [10,11].

Despite the challenges and barriers, attempts have been made to teach farmers about the unique and complex sector of sustainable agriculture. There are several currently known attributes and methods that lead to success in sustainable agricultural training: (i) Experiential methods of learning, which refers to combining experiential and information-focused teaching methods, instead of solely focusing on the latter. Currently, most programs focus on fact-heavy, teacher-centered techniques while neglecting the practices that behavioral and sustainability scholars highlight as central to creating change [14,15]. Learning through

experience has been deemed very useful, when particular attention was drawn to the application of knowledge in field settings, such as on-farm experiences including internships, student farms, short-term visits, and conversations with farmers [16]. The importance of experiential learning is further evident by the success of farmer field schools teaching integrated pest management (IPM), an element of sustainable agriculture, in India [17]. (ii) Participatory methods of learning, which places responsibility on students, encourages autonomous learning, prepares them to better deal with future uncertainties and complexities, and promotes lifelong education [18]. It may likely come across resistance and tensions from teachers that do not wish to change their educational approach [19]. For most people in education this is a major challenge, as we give up one type of power in the classroom and assume a larger role as catalysts in a meaningful and community learning landscape [18]. (iii) Design by user, in which farmers can be promoted to advance into sustainable agriculture, to involve them, and give them influence over the educational program [20,21]. Mistakes from past programs that discouraged farmers from acting sustainably can be avoided when farmers have influence on the shape and creation of an educational program. This is achieved through the so-called stakeholder effect, in which farmers strongly influence the staffing, content, instruction, evaluation, and composition of a planning group [22]. (iv) Co-learning, being the collaboration between farmers and scientists to create suitable sustainable knowledge adjusted to their local environment [23,24]. To realize sustainability in agriculture, it is important to relate knowledge to specific local social and spatial environments [25,26]. (v) Non-traditional curricula, academics in sustainable agriculture are challenging the status quo of agricultural training. Sustainable agricultural education requires progressive, integrated, experiential, interdisciplinary, and system-based curricula where learning grounds theory to practice in purposeful, relevant, social, and environmental contexts [16,27]. It is, therefore, important to include social and political sciences in the curricula, not only environmental and economical sciences (as in traditional agricultural training). Non-traditional curricula also demand the educational program be built differently. To prevent a program from focusing on fact-heavy and teacher-centered techniques, it should not neglect the behavioral and sustainable practices that create change. To address this, Redman (2013) proposes combining and using educational pedagogy (based on real-world learning, critical problem-solving, and experiential learning), behavior change (based on declarative, procedural, effectiveness, and social knowledge), and sustainability competencies (based on systems thinking and an understanding of interconnectedness, the long term, foresighted reasoning and strategizing, stakeholder engagement and group collaboration, and action orientation and change-agent skills) as the basis of the program [14]. (vi) Peer learning or farmers learning from and with each other. This is usually carried out in the form of sharing advice, feedback, and thoughtful questions about a new farming technique, a new technology, or knowledge. Farmers are open to and value the practice of peer learning [28] as one of farmers' most commonly cited sources of information and ideas are other farmers [20,23]. Another reason why peer learning works is the fact that one farmer can be more knowledgeable on a certain topic, but can still learn through explaining, listening, discussing and working together with the other, who might be more knowledgeable on another topic [28]. (vii) Virtual education—in recent decades, education has undergone a tremendous evolution with the rise in new technologies in school settings. In agricultural education, this gave rise to the widespread use of computer-based technologies such as virtual reality. Virtual reality is a technology that "uses computer-generated simulation of a three-dimensional image or environment that can be interacted with in a seemingly real or physical way by a person using special electronic equipment, such as a helmet with a screen inside or gloves fitted with sensors" [29]. The use of virtual reality in an agricultural-education context is anticipated to grow in popularity [30]. According to Wells et al. (2020), many teachers believed their students could be effectively engaged in the learning process using virtual-reality technology as an instructional medium [31].

In this paper, we apply this question of training in sustainable agriculture to training in intensive greenhouse horticulture. The protected horticulture in Almeria is located in the south-east of Spain and is produced in what is considered to be the largest concentration of greenhouses in the world. It is also one of the main greenhouse horticulture regions of Europe (Figure 1), occupying an area about 35,000 hectares. This agriculture activity involves 15,000 farmers engaged in production activity, and 40,000 additional jobs, which leads to economic values between primary production and auxiliary industry of over EUR 3400 million. Its development started in 1960 and resulted in important economic and social benefits for the region, but also created social challenges and had negative impacts on natural biodiversity and resources, as well as creating social challenges [32,33]. Castro et al. (2019) and Sayadi et al. (2019) identified major challenges that sustainable greenhouse horticulture faces in Almeria [34,35]. These challenges were focused on governance of shared responsibility, sustainable use of water, biodiversity, circular economy, and image and identity. Another challenge to the sustainable greenhouse horticulture in Almeria focuses on technology and knowledge transfer. It is believed that the region's horticulture is highly dependent on adequate technologies, yet it lags in adoption. Most of these technologies employ digitalization and data to help producers make decisions and to make production of the highest quality and efficiency. To successfully implement and adopt this new agricultural technology and knowledge, there is a need for effective and efficient interactions between various partners in the field [36]. The need for training in sustainable agriculture and the precision agriculture technologies of farmers in Mediterranean greenhouses has also been identified in Greece [37].

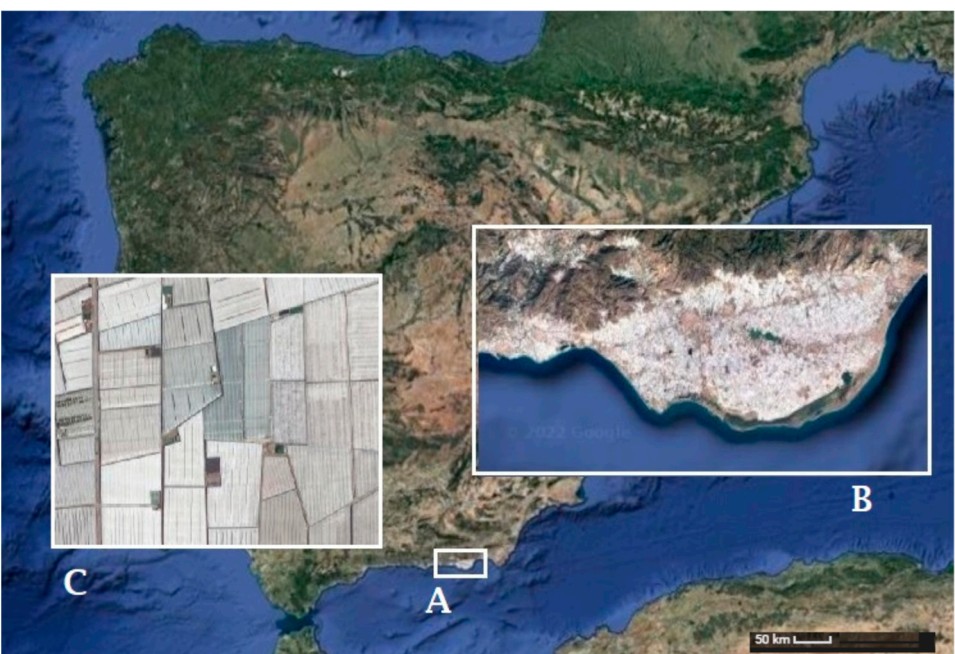

**Figure 1.** Part of the greenhouse horticulture area (**A**) in Spain. Detail of the area (**B**) and greenhouses (**C**) (Google Maps).

To tackle this challenge, the present research will investigate the best pedagogical practices and methods to transfer technology and knowledge, with the goal of improving the sustainability of greenhouse horticulture in the south-east of Spain, and in the Mediterranean region. The question posed in this paper is how to best train farmers in sustainable greenhouse horticulture, with particular attention drawn to pedagogical methods. In order to best address this need, information was gathered about the best pedagogical methods to use in (sustainable) agricultural training, how training could best be evaluated for future improvements, the barriers to adopting sustainable agriculture/horticulture, the necessary resources, and what steps should be taken to facilitate this new form of training. From this

information, a strategy to develop sustainable-greenhouse-horticulture training (the main objective of this research) could be determined.

## 2. Materials and Methods

The research was performed based on the question: "How can the pedagogical methods of training in sustainability be improved in the Almeria greenhouse-horticulture sector?", and developed through the following sub-questions:

1. How do different stakeholders understand sustainability in greenhouse horticulture?
2. What pedagogical methods are currently used for sustainable training in the Almeria greenhouse-horticulture sector?
3. How has the COVID-19 pandemic effected greenhouse-horticulture training in Almeria?
4. How do different stakeholders evaluate the existing sustainable training methods?
5. Which pedagogical methods are the most effective in training sustainable greenhouse horticulture?
6. What human, financial, physical, and information resources does an organization need to provide for training in sustainable greenhouse horticulture?

Sub-question 1 checked whether all stakeholders have the same definition of sustainable agriculture/greenhouse horticulture. The 2nd sub-question enabled the development of advice for change and improvements to the pedagogical methods currently used. During the internship, it became apparent that COVID-19 had influenced the greenhouse horticulture in Almeria. Therefore, sub-question 3 attempted to explore the effects of this phenomenon further. The 4th sub-question aimed to find out how experts evaluated the pedagogical methods currently used in Almeria's greenhouse-horticulture sector. Coinciding with this, sub-question 5 aimed to identify the best pedagogical methods to use and sub-question 6 discussed the exact resources needed to facilitate training in sustainable greenhouse horticulture.

To answer the main and sub-questions, there was a need for more than one method of data collection. A combination of an online questionnaire, a Delphi method, and desk research, each with an individual sampling frame and analysis method, was, therefore, used as the strategy to collect the data and implement the research design. The online questionnaire was utilized to answer sub-question 2 and 3. Sub-question 4 was answered by the Delphi questionnaire, and sub-questions 1, 5, and 6 were answered through a combination of desk research and the Delphi questionnaire.

(a) Online questionnaire

The first data-collection method was a questionnaire that aimed to identify how (sustainable) greenhouse-horticulture training was currently provided in Almeria. It had a study population and sampling frame of 'organizations providing (sustainable) greenhouse-horticulture training in Almeria'. The questionnaire consisted of 10 questions that were developed from the desk research, the sub-questions of this research, as well as the theoretical framework (Appendix A). They covered the topics of training, how a training comes to existence, the training target groups, the pedagogical methods used, evaluation of training, the use of protocols, how COVID-19 has effected the training sessions, and whether research has been carried out in what pedagogical methods to use.

A multiple-choice format was created for most questions, instead of open questions, in order to ensure a satisfactory sampling rate. It was still possible for a participant to give an elaborate answer, by selecting the additional option of 'others', if they deemed it necessary. Open questions were only formulated when it was absolutely necessary. The online questionnaire was designed this way to allow for easy answering, to avoid ambiguous answering, and to minimize the time constraint for participants. On the one hand, this resulted in the questions being of more general nature, but on the other hand, it resulted in a satisfactory sampling rate (see 'Study population and sampling' section) (Figure S1).

The overall aim of this questionnaire was to specifically answer sub-questions 2 and 3, and to develop a good overview of how training is provided in the whole greenhouse-horticulture sector of Almeria. A pilot test was performed initially, and the final survey was executed between 29 March and 14 April 2021.

(b)    Delphi method

The second and main method of research was a Delphi-method questionnaire. The Delphi method is a scientific way to engage the opinion of experienced experts through interview and discussion procedure among mutually anonymous participants in two or more cycles. The data from each cycle are processed and submitted to the participants for their further consideration and evaluation. This enables the collection of diverse data, ideas and opinions based on which we can, using consensus, define the terms, assume the events and/or establish a process flow or develop guidelines of action [38,39]. In the field of pedagogy, the Delphi method is mainly used to review the effectiveness of the approaches so far, to better understand a research subject, and to project possible changes [38]. The Delphi method also allows for flexibility, which is particularly important when the participants involved are busy professionals. Lastly, the Delphi method provides a viable tool for learning as much as possible from highly experienced practitioners in the least amount of time [40]. Based on the relevance of these factors to the nature of this research, the Delphi method was chosen as the most suitable method for answering the research questions and achieving the research goal. In the context of this research, the Delphi method was performed following a set of 19 steps (Figure 2).

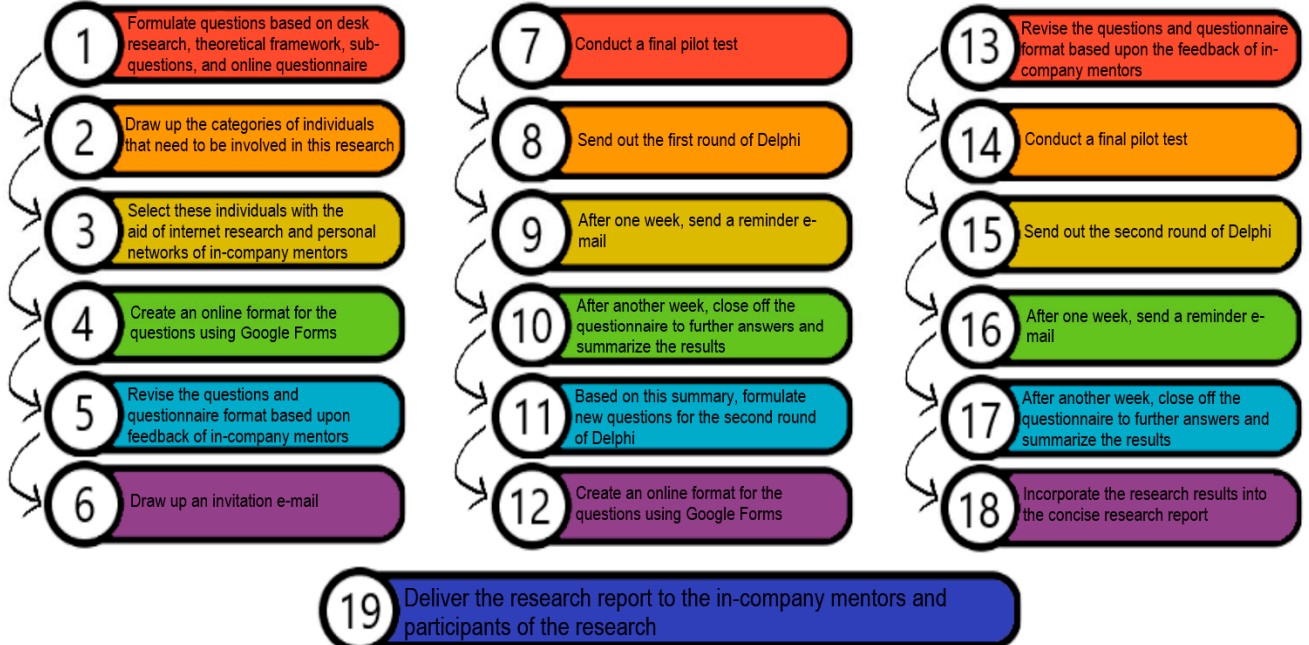

**Figure 2.** Steps of the Delphi method in the context of this research.

The Delphi questionnaire consisted of 2 rounds with 35 participants in the first round, performed from 21 April to 4 May 2021, and 27 in the second round, from 7 May to 20 May 2021. Google Forms was chosen as the tool to develop the Delphi-method questionnaire and collect its data. The first round consisted of 10 questions and covered the following topics: barriers to adopting sustainable agriculture/horticulture, personal definition of sustainable agriculture, the importance of several pedagogical methods, what would have to change in the current situation of greenhouse horticulture in Almeria, and what would be necessary to facilitate that change. These questions were derived from the desk research, the sub-questions of this research, and the results of the online questionnaire. The results of this round were analyzed and summarized with the affinity method (see below). The complete

structure of this round can be viewed in Figure S2. The second round also consisted of 10 questions. These questions were based on the summarized results of the first round with the goal of finding consensus among the experts involved. This consensus formed the basis for development of advice and guidelines of action in the advisory report. The results of this round were analyzed and summarized with the affinity method. The results of this round are discussed below, and the structure can be viewed in Figure S3.

(c)   Desk research

The last method of data collection was desk research, which formed the bases of the first two research methods and complemented them. Academic and non-academic literature sources were used to gain knowledge on the following topics: the best pedagogical methods for (sustainable) agricultural training, the barriers to adopting sustainable agricultural practices, and the specific challenges Almeria's greenhouse-horticulture sector faces in light of achieving sustainable production.

There were two study populations and sampling frames that each had their own method of data collection. The study population for the online questionnaire was organizations that operate in Almeria's greenhouse-horticulture sector, with the sampling frame consisting of organizations that provide training in Almeria's greenhouse-horticulture sector. All major organizations providing greenhouse-horticulture training in Almeria were identified and invited to complete the online questionnaire. This amounted to 12 individuals, spread across the 10 major organizations providing greenhouse-horticulture training. The final response rate was 11 out of the 12 invitees; their personal names are not mentioned, as they were promised anonymity. The study population of the Delphi method was more diverse and included greenhouse horticulturalists, technical advisors, organizations providing greenhouse-horticulture training, scientists, pedagogical experts, and trainers in sustainable agriculture. Individuals in these categories were identified through the aid of internet research (YouTube, academic articles, and website) and personal networks. The communication was performed through phone calls, e-mails, and visits. Each category of experts had to fit their respective sampling frame, which can be seen in Table 1.

**Table 1.** Definition and requirements of each expert category.

| Type of Stakeholder | Requirements/Definition |
| --- | --- |
| Main providers of training in Almeria | An organisation providing training in Almeria's greenhouse-horticulture sector. |
| Horticulturalists in Almeria | Horticulturalists that are representative of the majority of the population in terms of farm practices used. This was deemed more valid, as the majority of horticulturalists in Almeria do not (yet) have a focus on sustainable production. |
| Trainers of sustainable agriculture | Organizations or individuals that specifically train farmers in sustainable agriculture, as practices in training sustainable agriculture overlap with training sustainable greenhouse horticulture. |
| Scientists | Individuals that create new knowledge, techniques, and technologies in the field of greenhouse horticulture. |
| Pedagogical experts | Experts in the best (pedagogical) methods to use in (sustainable) agriculture and/or horticulture training. |
| Technical advisors in Almeria | Individuals active as advisors to, and quality controllers of, horticulturalists, as every farm has 2 or 3 advisors coming every week to check up on the plant health. These are quality rules, set by the government and/or cooperatives that buy their products, that every farm has to adhere to. |

The initial aim was to engage a minimum of 12 representatives, to achieve balance and data saturation. This was later increased to a minimum of 20, due to the variety of experts in these categories. This initial aim, the adapted aim, the number of individuals invited for each category, and the final response rate to both rounds of Delphi can be seen in Table 2.

**Table 2.** Sampling rate of the Delphi in the context of this research.

| Type of Stakeholder | Initial Aim | Adapted Aim | Invitees | Respondents First Round of Delphi | Respondents Second Round of Delphi |
|---|---|---|---|---|---|
| Main providers of training in Almeria | 2 | 8 | 13 | 7 | 7 |
| Horticulturalists in Almeria | 2 | 2 | 7 | 7 | 4 |
| Trainers of sustainable agriculture | 2 | 2 | 10 | 4 | 4 |
| Scientists | 2 | 4 | 10 | 6 | 5 |
| Pedagogical experts | 2 | 2 | 13 | 6 | 5 |
| Technical advisors in Almeria | 2 | 2 | 10 | 5 | 2 |
| Total: | 12 | 20 | 63 | 35 | 27 |

The aim was to engage 20 experts that would sufficiently represent the 6 categories. A total of 63 individuals were invited, of which 27 engaged in both Delphi rounds, thus satisfying this aim. The other important factor, that of sufficient representation of the 6 categories, was, similarly, achieved.

The online questionnaire consisted of open and closed questions. The closed questions were summarized by Google Forms and formatted into diagrams. The open questions were analysed with the help of the affinity method [41]. With this method, the answers were organised into groups or themes based on their similarity and then formatted into diagrams. These diagrams were used as visuals in the Delphi and results of this research. An example of the affinity method utilized in this research is shown in Figure S4. In case of the Delphi-method questionnaire, the first round consisted of open, closed, ranking, and selection (choose the top 5) questions. The closed, ranking, and selection questions were summarized using Google Forms and formatted into diagrams. The open questions were analysed with the help of the affinity method, organised into groups based on their similarity, and then formatted into diagrams. These diagrams were then used to display the results from the first round into the second round and as a basis to develop follow-up questions. The second round consisted of open, closed, and ranking questions. The closed and ranking questions were summarized using Google Forms and formatted into diagrams. The open questions were again analysed with the help of the affinity method, organised into groups based on their similarity, and then formatted into diagrams.

The best pedagogical methods, barriers to adopting sustainable agriculture/ horticulture, and the specific challenges Almeria's greenhouse-horticulture sector faces in light of achieving sustainable production were identified in the desk research and compared to the results from the Delphi-method questionnaire. If they matched, they were integrated into the professional product and the research results section of this report to complement the primary data. If the information collected in the desk research contradicted the results of the research, an explanation was given and included in the research results.

To answer the central question, and simultaneously ensure validity, this research made use of triangulation. This was performed through the combination of secondary data collection (literature and video sources) and primary data collection (Delphi method and questionnaire). Through gathering data and expertise from multiple sources and multiple categories of experts, validity was ensured. The validity of the Delphi method was ensured by basing its protocol and structure on the desk research, results from the online questionnaire, and the theoretical framework. This outlined the formulated questions, the general procedure of the Delphi method, and how respondents were to be approached.

For the Delphi method, a set of criteria was created for each group of experts. With this set of criteria, each individual could be categorised in a specific group and it could be checked whether the individual adhered to the description (and, thus, was of value to the research). An aim was set, and achieved, for the quantity of participants and sufficient representation of all expert categories. Furthermore, peer debriefing was extensively used

on the content of both questionnaires. The online questionnaire was revised three times; both rounds of Delphi were revised twice.

## 3. Results

The results are presented following the structure of the sub-questions and based on the theoretical framework (Appendix A). Some quotes marked with quotations ("—") are taken directly from answers provided in the online questionnaire or Delphi. These quotes are not referenced directly to an individual and instead to their relevant category group, so as to ensure the promised anonymity (see Figure S5). In the results, sustainable agriculture describes the umbrella term under which sustainable horticulture falls. This means that whenever sustainable agriculture is mentioned, it implies that these results can be applied to facilitate sustainable horticulture as well.

### 3.1. The Definition of Sustainable Agriculture

The first sub-question aimed to find out how sustainable agriculture was understood and defined by different stakeholders. Experts identified the following six principles to which agriculture must adhere for it to be sustainable and realistic to adopt (Table 3):

**Table 3.** Definitions of sustainable agriculture.

| |
|---|
| (1) Respects people |
| (2) Profitability |
| (3) Respects the planet |
| (4) Uses natural and non-natural resources efficiently |
| (5) Improves the health of the land |
| (6) Meets the current needs while not jeopardizing production for future generations |

'Respects people (1)' emphasizes the well-being and livelihood of people directly and indirectly involved in sustainable agriculture. Examples of this include: work not being overlaboring, not using materials that are hazardous to human health, and the creation of healthy agricultural products. 'Profitability (2)' means ensuring a fair price for the products produced that not only covers the costs, but also allows for future investments.

Agriculture in any of its forms will end up damaging the health of the earth. It must always be considered that an agro-system itself is a system that has been manipulated by man instead of nature. To achieve 'Respecting the planet (3)', the negative consequences derived from these agricultural practices must, therefore, always be minimized. The 'Efficient use of natural and non-natural resources (4)' emphasizes the use of circular principles. Waste and pollution should be designed out, products and materials should be kept in use for as long as possible, and natural systems should be regenerated (Scientist 3). When there does occur a need to use non-natural resources, such as chemical pesticides, this should be performed in a calculated manner. Science, biology, and technology could be used to determine the exact quantity needed to maximize the efficiency of non-natural resources.

"A sustainable agricultural practice should concern itself with the health of the cultivated soil, trying to maintain and/or achieve optimal health (Sustainable agricultural trainer 2)". 'Improves the health of the land (5)' thus means that there will be cases where health can be improved and other cases where you only maintain the health already achieved. Sustainable agricultural practices are defined by long-term sustainable use of agricultural land. The generations that come after us should not suffer the consequences of our actions. Besides this, it is important to maintain an efficiency of the agricultural land that still meets the needs of the current population. A balance should, therefore, be maintained which "meets the current needs, while not jeopardizing agricultural production for future generations (6)".

### 3.2. Training in the Greenhouse-Horticulture Sector of Almeria

This research was situated in the greenhouse-horticulture sector of Almeria. For that reason, effort was made to find out how (sustainable) greenhouse-horticulture training worked, and is provided, in Almeria. The online questionnaire was the primary research method for this section. The second sub-question aimed to find out which pedagogical methods are currently used for (sustainable) greenhouse-horticulture training in Almeria. The major methods used are on-farm/business demonstrations, virtual education, and classroom education (Figure 3). In the context of Almeria's sector: on-farm/business demonstrations are where new cultivation techniques and technology (among other things) are demonstrated in real, or closely simulated, greenhouses; virtual education covers online conferences in which research results are discussed and training sessions provided; and classroom education is the traditional pedagogy where teachers are at the center of information, usually used for the explanation of theoretical material. As of today, no research has ever been conducted by any of the major organizations in Almeria regarding what pedagogical methods to use in their training activities.

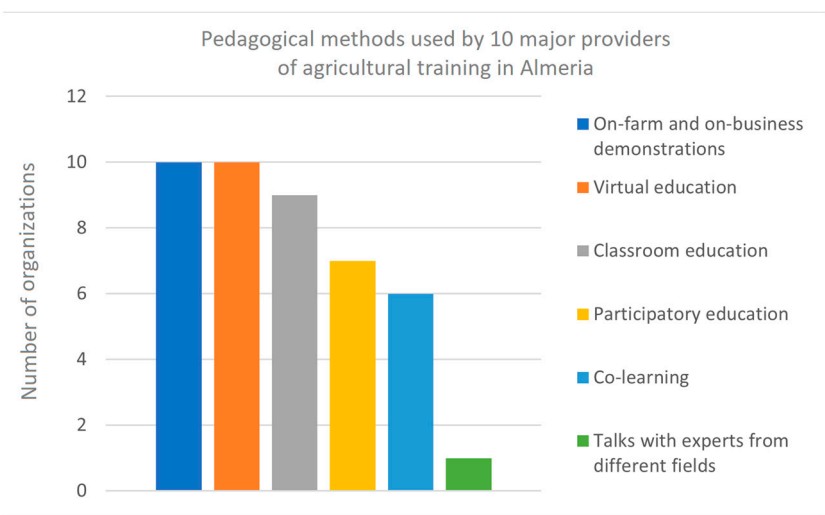

**Figure 3.** Pedagogical methods used by training organizations in Almeria.

Over the years, the training in Almeria has been continuously developed. Most of the organizations providing greenhouse-horticulture training in Almeria employ scientists and engineers that conduct research and develop technology. As a result of this, the content of training activities have been continuously updated and adapted in various ways. Most notable has been the increasing use of virtual education: online platforms have been created where research results and consultation are shared; educational apps have been developed; and YouTube is regularly used to share new content.

Surprisingly, the largest target audience of the greenhouse-horticulture training activities in Almeria were academic agriculture students, and, thus, future agriculture professionals (Figure 4). The online questionnaire discovered that there was quite a diversification of training topics provided by the organizations (Figure 5). It was somewhat surprising that only one organization provided training in food quality and safety. The most common themes discussing sustainability were biological pest control (e.g., the use of insects to combat pests) and decreasing the use of fertilizers. Both these themes showed promising results and did combat the major problem in Almeria's greenhouse-horticulture sector: water and ground pollution.

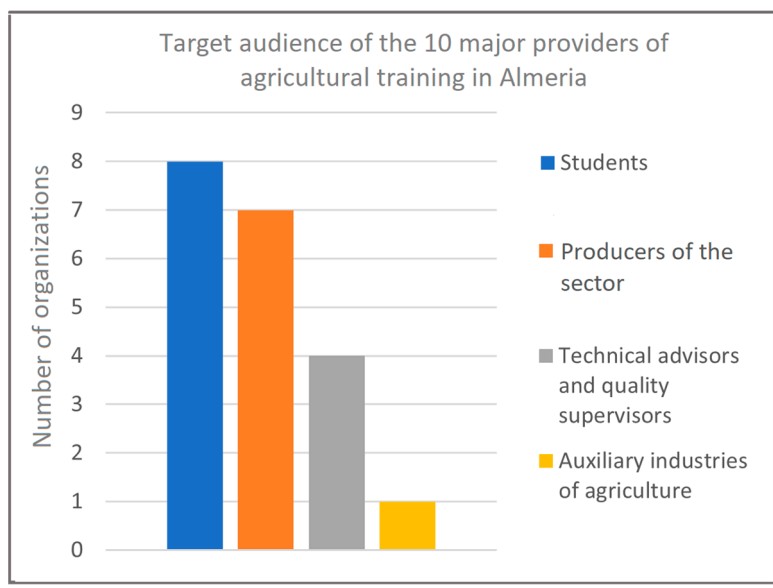

**Figure 4.** Target audience of training.

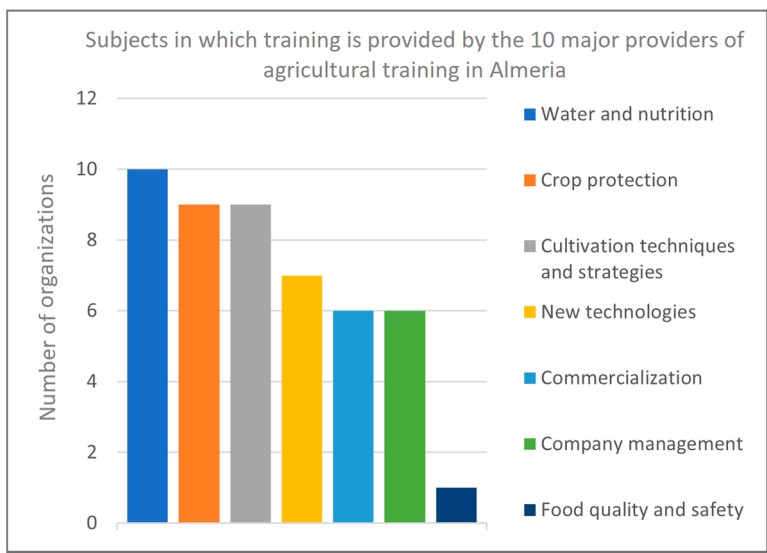

**Figure 5.** Areas and themes of training.

Training mostly originated from market demand and the own initiative of training organizations in Almeria (Figure 6). The main reason for the 'own initiative' is that these organizations often had more information on current and future events compared to farmers. For example, one of the organizations was more likely to know if there would be any bans or regulation changes in the near future (Training organization 6). With this knowledge, the organizations would take their own initiative and decide that it is important for farmers and students become familiar with substitutes through training activities.

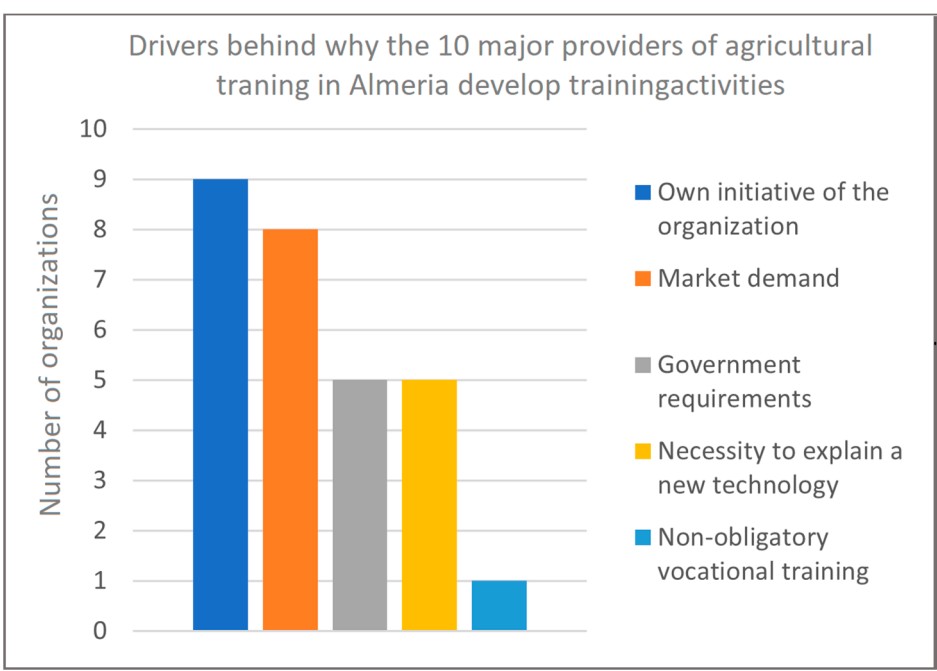

**Figure 6.** Origins of training activities.

### 3.3. The Impact of the COVID-19 Pandemic

The third sub-question aimed to find out how the COVID-19 pandemic had affected greenhouse-horticulture training in Almeria. In general, both theoretical and practical training was adjusted to virtual education. Initially, all training was postponed or stopped completely. After a while, most on-farm/business were replaced with video conferences. This is the main reason why there had been a stark increase in the use of virtual education. This forced use of digital technology brought a positive development to several organizations, as it resulted in improvements in their teaching. Despite the innovative adaptions, most organizations struggled to effectively reach their target audience and to carry out tailor-made courses. Whenever there were non-virtual on-farm/business demonstrations (at later stages of the pandemic), the logistics were adapted to meet health security measures such as sanitation stations and sufficient distance between visitors.

### 3.4. Evaluation of the Pedagogical Methods Currently Used

The fourth sub-question aimed to discover how different stakeholders evaluated the existing sustainable training methods used in Almeria's greenhouse-horticulture sector. Specific advice given by the expert panel was to increase the talks with experts from different fields, co-learning, and participatory education (Figure 7). The emphases on on-farm demonstrations (OFDs) should remain the same while the use of virtual education should be reduced, after the COVID-19 pandemic was over. Furthermore, the expert panel advised to not just stick to one method, but instead to combine several methods to teach the content of a training activity. Further information about the best pedagogical methods to use can be found below.

Some interesting information was provided by the expert panel regarding talks with experts. They believed that there had been such an excess of information (or misinformation), that the possibility of sharing experiences and fears with experts and peers helps a lot to filter what is useful and truthful, and what is not. It is very important to provide technical and detailed information on the sustainable practices promoted; it is also of special interest to have technical information adapted to different casuistry and cultivation realities. "Talk with pioneer farmers, the 'good examples', and field technicians with experience in the sustainable practices" was a frequent and important learning method (Technical advisor 2'). These expert talks were often combined with on-farm/business

demonstrations, again emphasizing the advice to combine several pedagogical methods to facilitate a training activity.

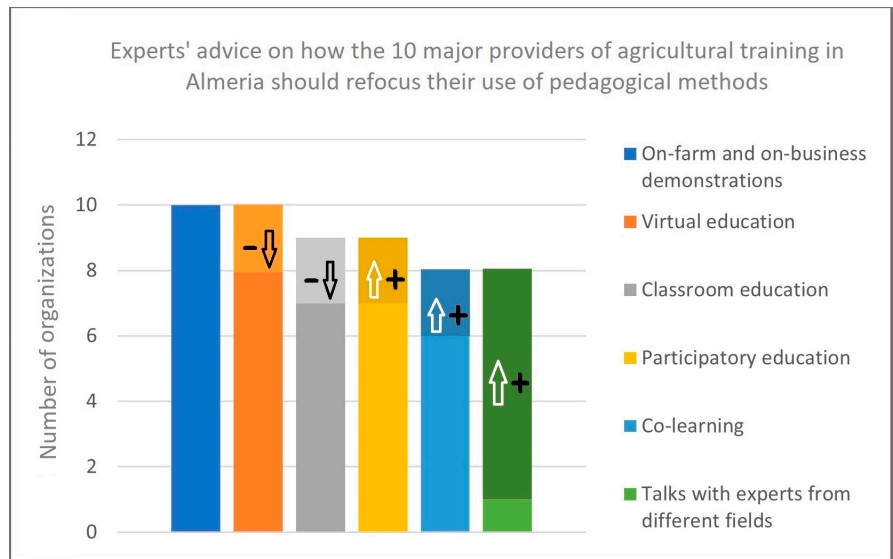

**Figure 7.** Expert advice on the current pedagogical methods used during the study and suggested changes (positive and negative arrows).

*3.5. The Best Pedagogical Methods to Use in Sustainable Agricultural Training*

The fifth sub-question gathered expert advice and insights on what pedagogical methods to use in sustainable greenhouse-horticulture training (Table 4). On-farm/business demonstrations, participatory learning, and co-learning were deemed as the most effective pedagogical methods to use in sustainable agricultural training.

**Table 4.** Pedagogical-methods averages ranked based on their effectiveness.

| Pedagogical Method | Average Values Ranked from 0 to 9 |
|---|---|
| On-farm/business demonstrations | 8.7 |
| Participatory learning | 8.6 |
| Co-learning | 8.4 |
| Peer learning | 7.9 |
| Design by user | 7.4 |
| Holistic/non-traditional curricula | 7.0 |
| Virtual education | 6.0 |
| Traditional classroom training | 5.2 |

Peer learning was ranked lower for its complexity, especially in managing the peers well to create an effective training environment. Pedagogical expert 3 mentioned that peer learning, at its core, is the combination of on-farm/business demonstrations, participatory learning, and co-learning. This combination makes peer learning difficult to realize yet, at the same time, a highly effective method for agricultural training; therefore, it was ranked relatively lower by the experts. Design by user was deemed as important, but simply not as important as the first-four pedagogical methods and was, therefore, ranked lower. Holistic/non-traditional curricula was, on average, ranked with a 7.0 as the experts sometimes deemed this method to be impractical. The experts acknowledge the importance of discussing sustainable agriculture in a universal context but are fearful of keeping training activities too general and lacking the necessary specifics. "Virtual education is great for reaching a wider audience but misses social interaction that e.g., on-farm/business demonstrations and participatory learning can provide" (Trainer of sustainable agriculture 4). This method can be good in terms of time efficiency for farmers that are not able to leave

their farm obligations. However, the experts regarded virtual education as a complementary method, not as a main method for teaching sustainable agriculture. Traditional classroom training received the lowest score. The experts recognize that the best way to learn about sustainable agriculture is in the field, which makes traditional classroom training an unsuitable pedagogical method to use. "At best traditional classroom education could be utilized to teach materials of purely theoretical nature" (Scientist 3). The most prominent advice given by the expert panel was for training organizations in Almeria to consider selecting and combining several pedagogical methods to teach a training activity. The mentioned pedagogical methods can be combined, and complement each other, in order to create the best manner of training for every occasion and theme. To implement these pedagogical methods on a consistent basis, there is a need for protocols. "It's crucial to know why you are including specific parts of a training and to know what the sub-goals are" (Pedagogical expert 1). The expert panel advised organizations to use specific protocols for all the offered training (96%), and some believed that a general protocol is essential for success (4%) (results not shown). An important factor in maintaining the quality of training programs is evaluation. The expert panel concluded that the most feasible way to assess whether short-term and long-term learning outcomes have been achieved is by measuring indicators before and after training to assess its adoption and impact. "These indicators should be both of qualitative and quantitative nature, not just asking 'how many courses have you attended'" (Technical advisor 2). It would be even better to investigate changes and adaption levels over a longer period of time (2–3 years), when more resources are available.

### 3.6. The Resources Needed to Facilitate Sustainable Agricultural Training

The last sub-question discussed the human, financial, physical, and information resources that an organization needs to provide training in sustainable greenhouse horticulture. The human resources refer to the skills and expertise of employees. Financial resources refer to the available finances, and their flow, in an organization. Physical resources refer to tangible resources such as buildings, land, machinery, and equipment. Information resources refers to the data that an organization can use to guide its operations. The expert panel concluded the following resources are important for success in realizing sustainable-agriculture training (Table 5):

**Table 5.** Resources needed to provide sustainable agricultural training.

| Specific Resource | Type of Resource |
| --- | --- |
| Suitable trainers | Human, financial, and information |
| Updated and relevant training programs | Information and human |
| Collaboration between stakeholders of the entire sector | Human, information, and financial |
| Funding | Financial |
| Demonstrations on farm where sustainable practices have successfully been adopted | Physical, financial, human, and information |

Suitable trainers are needed to specifically train practices of sustainable agriculture. To realize this, there is a need to train and expand the skill set of specific employees (human resources) regarding the process of sustainable-agriculture training discussed in this research, if this is not done so already. "This requires organizations in Almeria to expend the scientific information resources to include research in pedagogy" (Pedagogical expert 2). Further financial resources are needed to cover the expenses of this transition. Training programs have to be relevant and constantly updating in the rapidly growing industry of sustainable agriculture. There is a need to develop new technology and cultivation techniques suitable for sustainable agriculture. This process is already realized in Almeria (Figure 5), as all organizations have a research department for this purpose. The difference here is that in Almeria, these departments are not always focused on sustainable-agriculture research. To realize this, there is a need for human resources in the form of capable scien-

tists and experts in, for example, economics and pedagogy. These individuals support the development of information that can achieve a change in current agriculture towards a more sustainable one.

Technical advisor 1 mentioned that another important resource in realizing sustainable-agriculture training was a collaboration between whole sector (farmers, students, technicians, researchers, etc.). Information and human resources should be pulled together, especially from the main providers of training in Almeria, in order to advance sustainable agriculture. This collaboration does come at a financial cost, as some of these organizations are competitors and may, therefore, need to be persuaded with financial resources to share their expertise and information. "Human resources are also needed here, as there will be a need for specially skilled employees to guide and manage a network of this size" (Main provider of training in Almeria 2). Financial resources in the form of funding were also found to be crucial for success in sustainable agricultural training. Examples of funding may include the invitation of international experts, financing research programs, financing training programs to smoothly transfer research information, financial support for pioneer farmers that test new concepts in their greenhouses, and the purchase of equipment (physical resources).

The last resource mentioned by experts is the farm demonstrations where sustainable practices have successfully been applied. For this to be realized, all resources would have to be combined: there is a need for human resources in the form of skilled individuals that guide the demonstrations; a need for information resources in the form of scientific evidence that the sustainable practice demonstrated works well; a need for physical resources in the form of farms and land where sustainable practices can be tested and presented; and a need for financial resources to fund the complete process.

### 3.7. Main Reasons Why Farmers Do Not Adopt Sustainable Agriculture

The main reasons that limit farmers from adopting sustainable agriculture are: perception of high costs to implement sustainability (listed by 21 out of the 35 experts); lack of tests and demonstrations on the farm (18/35); low perceived effectiveness of sustainable-agriculture practices (17/35); resistance to change (18/35); and lack of training and information on sustainable agriculture (22/35). These reasons should be considered when designing a training activity. "Regular check-ups should be conducted to confirm that these reasons are not blocking farmers from adopting sustainable practices" (Pedagogical expert 3).

### 3.8. Most Common Mistakes Made in Sustainable-Agriculture Training Programs

Finally, the research discussed the most common mistakes made in sustainable-agriculture training. These mistakes included: the idea that farmers do not receive help on a continuous basis in the process of adopting sustainable agriculture; an improper way of transferring information; the lack of practice opportunities; the lack of evidence that sustainable agriculture works; and the improper idea of what sustainable agriculture would be due to too much emphasis on one specific element. These mistakes were ranked according to their impact and frequency of occurrence (Table 6).

**Table 6.** The ranked numbers of experts agreeing with statements regarding the most common mistakes made in sustainable agricultural training.

| Common Mistakes | Ranked between 0 and 100 |
| --- | --- |
| Farmers do not receive the continuous help that is needed | 94 |
| Improper way of transferring information | 93 |
| Lack of practice opportunities | 88 |
| Lack of evidence that sustainable agriculture works | 86 |
| Too much emphasis on a specific element of sustainable agriculture | 80 |

Lack of practice opportunities is a common mistake made in sustainable agricultural training. In many cases, training programs only offer information on a theoretical level, including minimum (sometimes even absent) field work. These forms of training programs create a gap between what is portrayed in the classroom and the reality of applying sustainable practices to a farm. The lack of credibility of the topics discussed consequently creates an improper way of transferring information, another common mistake. "This often occurs when the scientific evidence and economic profitability of sustainable agriculture is inappropriately transferred" (Scientist 4). Another mistake made is the overemphasizing of one specific element of sustainable agriculture, thus portraying an incomplete idea of how sustainable agriculture works. Examples of this include: programs that merely teach sustainable agriculture according to regulations set by government agencies and/or companies; overemphasis on, for example, biological pest control while neglecting other practices, such as waste management or reduction in chemical fertilizers, which would make the practice of sustainable agriculture whole; and an overemphasis on environmental aspects while neglecting the quality, health, and economic advantages that sustainable agriculture can bring. It is very important to provide technical and detailed information when promoting the effectiveness of sustainable practices. "Training programs should strengthen the perception that sustainable agriculture generates improvement and health to the agrological systems and make them more resilient in the face of possible disturbances such as droughts or pests" (Trainer of sustainable agriculture 2). Demonstrations on the advantages of sustainable agriculture are crucial for increasing its adopting rate. All this will facilitate the correct development of crops and economic profitability, transmitting the premise that "the profit is also generated in the savings". The last most-common mistake made is not supporting farmers on a continuous basis in their adoption of sustainable practices. "Changing farming structures takes time and cannot be realized by one-time events or training programs" (Main provider of training in Almeria 7). Training programs often lack a form of follow-up and do not establish clear and measurable objectives for the farmers.

## 4. Discussion

Sustainable agriculture is not easily defined and has had a great variety of meanings so far [42]. The experts consulted through the Delphi method attributed this term with the list of significances shown in Table 1. Respect for people must not only be contemplated in sustainable agriculture but is a fundamental pillar in sustainable human development, satisfying the needs and improving the living conditions of the population, without losing the objective of achieving an increasingly full life [43]. Respect for the planet, including the improvement of its health and the efficient use of natural and non-natural resources, are included as an important part of the definition of sustainable agriculture, in line with other authors [44–46]. The economic sustainability of farms is a constant within the framework of Common Agricultural Policy in the countries of the European Union [47] and promoting One Health for food safety and security is a way to engage the next generation in entering employment and education in agriculture and the food system [48].

During the last decades, the horticultural production in Europe has gradually moved from the northern countries towards the Mediterranean basin. In the latter, it is concentrated in the south-east of Spain, within and surrounding the province of Almeria, and is based on low investments, and low dependency on energy [49]. The research in the present paper is situated in the greenhouse-horticulture sector in Almeria, and, for that reason, effort was made to find out how (sustainable) greenhouse-horticulture training works, and is provided, in this region. Thus, a question aimed to find out which pedagogical methods are currently used for sustainable greenhouse-horticulture training in Almeria. It showed that the major methods used are on-farm/business demonstrations, virtual education, and classroom education (Figure 3). On-farm/business demonstrations are where new cultivation techniques, technology and other issues are demonstrated in real, or closely simulated, greenhouses; virtual education covers online conferences in which

research results are discussed and training sessions provided; and classroom education is the traditional pedagogy where teachers are at the center of information, usually used for the explanation of theoretical material. Until today, no research has ever been conducted by any of the major organizations in Almeria on the pedagogical methods to be used in their training activities. Over the years, the training in Almeria has been continuously developed. Most of the organizations providing greenhouse-horticulture training employ scientists that conduct research and develop technology. As a result of this, the content of training activities is continuously updated and adapted in various ways. Most notably is the increasing use of virtual education: online platforms are created where research results and consultation are shared, educational apps are developed, and social media such as YouTube is regularly used to share new content.

The largest target audience of the greenhouse-horticulture training activities in Almeria are agriculture students (Figure 4). Most of these come from the regional university and seek applied knowledge and training. Nevertheless, there is also a fundamental proportion of young people who want to join the agricultural activity in the sectors of intensive agriculture. They receive a training course on the Incorporation in the Agricultural Business together with professionals who work in the agricultural and livestock sector but have no higher education and need to have a certificate of professional agricultural training. On this course, several modules are taught on the creation, management, organization and legislation of this type of business both at the state and European levels. Students learn about marketing, conservation of the environment, and accounting, etc. There is an important part aimed at protecting the environment through the use of resources, which also includes good waste-disposal practices, and ecological and integrated production.

The online questionnaire discovered that training topics are quite diversified (Figure 5). On the other hand, it was somewhat surprising that only one organization provided training in food quality and safety. This can be explained by the strict rules and regulation procedures that every farmer must already adhere to in Almeria. The most common themes discussing sustainability are biological pest control (e.g., the use of insects to combat pests) and decreased use of fertilizers. Both these themes show promising results for greater sustainability in Almeria's greenhouse-horticulture sector. All the themes are widely connected with the Sustainable Development Goals (SDGs) of the 2030 agenda and, achieving their fulfilment and advancing towards the achievement of sustainable development, requires a joint effort [50,51].

While the interviewees believed that the market demands determine the content of training and education (Figure 6), this is probably also linked to the major challenges of European agriculture: the need to produce high-quality food, and to meet increasing demands of nature conservation and environmental protection, and provide, at the same time, good and secure jobs. This has a clear political connotation where politicians have called for comparable and compatible vocational qualifications (as envisioned in the "Bologna Process"), training according to the principles of Education for Sustainable Development and, most recently, an action plan for digital education [52].

In food production, legislation for sustainability and new markets for ecological foods or other sustainable food-production methods go hand in hand. The proposal for a legislative framework for sustainable food systems (FSFs) is one of the flagship initiatives of the Farm to Fork Strategy [53]. As was publicized in the Strategy, it was to be approved by the Commission by the end of 2023. Its goal has been to accelerate and facilitate the change to sustainable farming systems. It will have fundamental goals which include to improve the consistency of EU and national policies, normalize the practice of sustainability in all food-related guidelines and reinforce the resilience of food systems. This also involved an open public consultation for the sustainable-food-system framework initiative that was published on 28 April 2022 and ran until July 2022. This public inquiry was to collect information and suggestions from pertinent public and private stakeholders (governmental authorities, social and economic organizations, research and

academic institutions, NGOs, and citizens in general) on the main topics of the initiative (https://ec.europa.eu/info/law/better-regulation/ (accessed on 1 August 2022).

The impact of the COVID-19 crisis on training has pushed the educational boards globally to re-examine their training curriculum [54]. Agriculture is one more area where there was a training break due to the unexpected and dangerous pandemic situation. The entities dedicated to agricultural training that previously had didactic materials and suitable platforms for virtual education, as was the case in Almeria horticulture, were quickly launched in online mode. Prior to the COVID-19 epidemic, didactic materials presented little evidence of innovative strategies and did not fully exploit the potential of information and communication technology (ICT) in the design and use of materials. With the new need to introduce online training, some of the simple didactic materials underwent little modification or transformation regarding printed materials. Other materials were of greater complexity both from a design point of view and in their pedagogical implications. Most materials were unknown to the teachers and demand significant participation in decision-making. However, in both cases, many of the resources had not been evaluated nor experienced, which created a significant degree of uncertainty in relation to their possibilities for didactic use [55].

Likewise, teacher resilience was the key to the success of this transformation [56]. Obviously, the computer equipment of farmers is an essential part of the realization of new methods of virtual training. In the province of Almería, most farmers have a single computer although some of them have more, allowing it to be available both at home and on the farm [57]. The videos had successfully replaced the visits that are often made for the explanation of new technologies on farms and, as in other disciplines, face-to-face activity in agricultural training has been organized once again following the pandemic with the appropriate precautionary measures. Therefore, the COVID-19 pandemic and the consequent confinement produced a substantial change in the teaching and learning process; the much-mentioned educational digitalization is, today, a reality that is here to stay [58].

When the experts were asked for the preferred participative training, they mentioned co-learning and talks by experts over classroom or virtual training (Figure 7, Table 4). New forms of education such as e-learning have become omnipresent and have contributed to a new educational model in general [59,60] as well as issues of sustainable development [61]. E-learning covers several concepts (blended learning, virtual learning, learning management systems) and are used interchangeably with technical notions (computer-based learning, online learning, technology-enhanced learning). New emerging trends such as massive open online courses (MOOCs), mobile learning, and digital learning have made it more difficult to distinguish between the different modes and make it harder to define them. From the perspective of sustainable development, e-learning initiatives should promote and improve continuing education, ensure the acquisition of sustainability-related knowledge and skills, and increase public awareness and understanding of sustainable development and its implications [62]. However, the ultimate value that users will obtain from e-learning tools will depend on the extent to which the target population effectively uses the technology [63]. Despite the increasing use of information systems in the agricultural sector [64] and the awareness of the benefits of ICT applications in farming activities [65], previous works have revealed the low acceptance and adoption of these technologies among farmers [66–70]. For this reason, understanding the aspects that move users to accept or reject these types of technologies is paramount [63]. In the meantime, it may explain why the experts in our study preferred other types of training.

Likewise, participative training, co-learning, and other activities such as on-farm/ business demonstrations (OFDs) were deemed as the most effective pedagogical methods to use in sustainable-agriculture training (Table 4). OFDs as agricultural-knowledge exchange activities [71] have already been identified as a good strategy to shift towards more partici- patory agricultural-education activities, since these events have the potential to facilitate dialogue between attending farmers and other interested parties, for example, researchers,

advisers and suppliers. Furthermore, it has been argued that improvement in using the full potential of OFDs as strong learning environments in Europe would be beneficial [72] and a step toward reaching the SDGs [51]. An OFD is a demonstration activity or event for providing farmers with 'an explanation, display, illustration, or experiment showing how something works' (Collins English Dictionary), which can be subsequently transferred to their own farming practices to bring about positive changes on their farm [73,74]. During an OFD, researchers, farmers themselves or others can take up the role of demonstrator. They occur, preferably, on actual working farms, or in a context which is as realistic as possible. Hence, the demonstration can be visualized in real-life conditions to which farmers can relate [72]. OFDs could, thus, be deployed for more traditional transfer-of-knowledge activities, but also for actively engaging bottom-up learning activities. These activities could include providing the opportunity for farmers and other attending interested parties to discuss together with both peers and experts, jointly solve problems, compare practices in similar contexts to their own, as well as experience hands-on activities [20,21,73,74].

In the present Delphi research, peer learning was ranked lower for its complexity, especially in managing the peers to create an effective training environment. Interestingly, many reports claim that farmers like to learn from peers who share reliable information [28,75–77] and that farm visits are one of the most preferred ways to bring information to farmers [23,24,75,78]. Thus, farmer-to-farmer learning should be in synergy with researcher–farmer learning experiences and learning groups consisting of different types of agricultural experts [79]. Agricultural experts would have a predominant role in the communication or co-creation of knowledge [71,80,81].

The fact that peer-to-peer learning was ranked lower also recalls earlier examples of reasons why the sharing of knowledge between farmers does not evidently occur naturally. Some farmers have been reported to become reluctant to engage in networks when confronted with new ways of working, due to fear of criticism from other farmers, of competition, or poor regard for the standards of farmers new to the system [82]. Many other studies on agricultural extension activities report facilitation as a key success factor [83–85], and OFDs as critical success factors [86,87]. The above-mentioned theory and empirical studies lead us to formulate the hypothesis that deliberately facilitating dialogue during OFDs between farmers and other attending parties increases stimulation of transformative learning processes, compared to OFDs during which dialogue is not deliberately facilitated.

In a recent bibliographic review on what determines decision making in agriculture, three types of behavioral factors were distinguished: dispositional, social and cognitive [88]. Dispositional factors are believed to be relatively stable to internal variables related to a given individual, such as personality, motivations, values, beliefs, general preferences and objectives [89]. Social factors relate to farmers' interactions with other individuals (e.g., other farmers or advisors) and include social norms; it involves what farmers perceive others expecting from them. Finally, cognitive behavioral factors are proximal and relate to learning and reasoning. Concerning sustainable agriculture, they include farmers' perceptions of the relative benefits, costs and risks associated with a particular sustainable practice or whether they feel that they are skilled enough to adopt this practice [88]. When asked for the motivation for adoption of sustainable agriculture, the panel of experts expressed two dispositional factors: one which is philosophical/ideological in nature, and another which is the environmental awareness of their farm and surroundings (proper ideology). A third motivation was because of market demands, which was emphasized as the most important of the reasons for adoption of sustainable agriculture, and should be considered a cognitive motivation factor. Thus, social factors were not cited as an important motivation. Instead, the panel proposed that farmers are motivated to practice sustainable agriculture because of requirements set by governmental agencies and/or companies, for example, the prohibition of the use of certain phytosanitary products, or the requisite to have certain certificates.

Farmers' decisions to comply with mandatory environmental regulations have been considered to be behavioral factors leading to complying or cheating and are different

from those leading to voluntary adoption [90]. Yet, this motivation argument should not be underestimated. A review of the literature showed that programs aimed at short-term economic benefits are rapidly adopted in contrast to those that focused only an ecological amenity. Sustainable practices can be adopted eventually by growers only when paybacks are perceived for either their farms, the environment or both [91,92].

When asked why farmers would not apply sustainable agriculture, the panel suggested several reasons such as the perception of high costs for implementing sustainability, lack of demonstrators on the farm, or lack of training and information on sustainable agriculture. As mentioned in 3.7., another reason was a resistance to change. This later reason has already been suggested for farmers not adopting more sustainable practices [93]. Resistance to change and personality are linked: individuals scoring low on receptiveness to new experiences may be particularly uneager to implement change in general [94]. The *status-quo* bias, whereby people systematically prefer to keep their current practices, is also intrinsically linked to resistance to change [95]. A recent meta-analysis on the role of the *status-quo* bias in agri-environmental policy showed that a high percentage of farmers systematically reject change [96]. A recent study in Greece suggested that elderly greenhouse farmers, with a low level of education, lack an innovation culture and show a distrust in innovative training. This is in contrast to young graduates that have returned to agriculture, and who are open to training activities and innovation [37]. As inertia is strong among farmers, it is probably one of the major reasons that more sustainable practices are not adopted [10,93].

The perception of high costs to implement sustainability is the mean reason and is in-line with the fact that economic factors were found to play a key role in farmers' willingness to participate in agri-environmental schemes [97]. Thus, public funding plans often fail to promote adoption due to lack of capital, inept strategy and unsuccessful directing of motivations. Growers recurrently fail to obtain the correct information about the profits of sustainable agriculture, as well [10]. The growers may resist change when they do not understand that sustainable farming can be necessary or beneficial and their proper experience may lead them to prefer using industrialized methods and crops, and believe that sustainable agriculture is relatively unproductive and challenging [98]. Many lack the general knowledge, practical experience, and skills required to effectively implement sustainable agriculture. Many potential but inexperienced farmers may be discouraged to start sustainable-agriculture practices due to a lack of the social networks, confidence, and resources that traditional growers have [10,99]. Traditional training and education organizations often fail to provide the correct information and training to growers because they tend to ignore social and ecological benefits [100,101]. Instead, they focus on short-term crop yields and productivity, and, here, farmers are seldom recognized as collaborative allies who can produce their own scientifically meaningful knowledge. In general, extension counsellors are not experienced in varied growing methods or participatory research. They are incompetent at supporting farmers, since their techniques and knowledge have not been established in actual farming situations, but under artificial extension-station conditions [101,102].

## 5. Conclusions

This research concludes that sustainable agriculture can be defined by the following characteristics:

1. Respects people;
2. Profitability;
3. Respects the planet;
4. Uses natural and non-natural resources efficiently;
5. Improves the health of the land;
6. Meets the current needs while not jeopardizing production for future generations.

In designing sustainable-agriculture training, one should avoid the following most common pitfalls:

1.  Farmers not receiving the continuous help that is needed;
2.  Improper way of transferring information;
3.  Lack of practice opportunities;
4.  Lack of evidence that sustainable agriculture works;
5.  Overemphasis on a specific element of sustainable agriculture.

This research concludes that in designing sustainable agricultural training, and achieving peer-to-peer learning, it is preferable to use:

1.  On-farm/business demonstrations;
2.  Participatory education;
3.  Co-learning.

In contrast to these methods, it is not preferable to use the following methods as the main pedagogy (although they can be used as complementary methods of training).

1.  Virtual education;
2.  Classroom education;
3.  Design by user;
4.  Holistic curricula.

The definitions of sustainable agriculture can now be used as the starting point from which sustainable-agriculture/horticulture training can be developed. When designing such training, the creator should make sure to target one of the mentioned principles. Above all, the designer needs to make sure to not present a training that contradicts these principles or that overemphasizes one principle and ignores the others, as this is a common mistake made in sustainable-agriculture training programs. With the definition of sustainable agriculture set, the designer has a set of principles to keep in mind and adhere to. This will support the designer in maximizing the effectiveness of—and avoid common mistakes made in—sustainable-agriculture training programs.

The most common pedagogical methods currently used in Almeria were identified, which showed the predominant use of on-farm/business demonstrations, virtual education, and classroom education. This research listed on-farm/business demonstrations, participatory education, and co-learning as the best pedagogical methods to use in sustainable-agriculture/horticulture training. A combination of these three pedagogical methods could realize effective peer-to-peer learning. However, due to the complexity of this pedagogical method it is ranked lower, and, therefore, is less advisable to immediately use as the dominant pedagogical method.

With this information, we can conclude that participatory education and co-learning should be further explored in the greenhouse-horticulture sector of Almeria, as they are proven to be successful pedagogical methods. Subsequently, we can conclude that virtual education and classroom education should play a less dominant role in the training activities. Instead, these pedagogical methods, together with design by user and holistic curricula, should be used as complementary methods.

To facilitate sustainable-agriculture/horticulture training, this research identified several essential resources. With these resources identified, organizations can now evaluate whether their training operations receive the right quality and quantity of resources.

The main thing learned from the effects of the COVID-19 pandemic is that virtual education gained a dominant role in the pedagogy of choice. Keeping the effects of COVID-19 in mind, this was the best option available. However, it is advisable for organizations to not stick with virtual education as their main pedagogical method, once the COVID-19 pandemic is over. Virtual education is a good complementary method, but ultimately fails to provide the practical experiences and social interaction that, e.g., on-farm/business demonstrations bring.

This research may have its limitations, though: it is heavily focused on the provider's side of sustainable agricultural training, and not as much on the receiver's side of training, as is evident by the relatively small group of actual farmers (horticulturalists) involved in this research. Although the experts involved are actively involved in the topic of this

research, there may still be an unforeseen difference between the conclusion of this research and the reality in the field of practice. Furthermore, the cost–benefit of pedagogical methods was not taken into account in this research. It may, for example, not be possible to utilize the advised pedagogy when limited finances and/or time is available.

The data gathered in this research is of a diverse and robust nature. The research was deliberately created to be of a robust nature, as no research had been performed in any Mediterranean greenhouse-horticulture sector regarding the best pedagogical methods to use in sustainable-agriculture/greenhouse horticulture training. This research identified the common mistakes made in sustainable-agriculture training programs, provided insights into why farmers adopt sustainable agriculture, and described barriers and (mis)conceptions that may prevent farmers from adopting sustainable agriculture/horticulture. This knowledge can help organizations and designers to avoid common mistakes, tailor their training activities, and be mindful of common barriers and (mis)conceptions.

**Supplementary Materials:** The following supporting information can be downloaded at: https://www.mdpi.com/article/10.3390/su15075816/s1, Figure S1: Online questionnaire (translated from Spanish to English); Figure S2: First-round Delphi questionnaire; Figure S3: Second-round Delphi questionnaire; Figure S4: Example of the affinity method utilized in the analysis of the online questionnaire and Delphi (example given is question 4 of the Delphi); Figure S5: Participants of the Delphi method.

**Author Contributions:** Conceptualization, R.D.W., S.S.G. and C.G.-G.; methodology, R.D.W. and S.S.G.; software, R.D.W. and D.J.; validation, R.D.W., D.J., S.S.G. and C.G.-G.; formal analysis, R.D.W.; investigation, R.D.W., D.J., S.S.G. and C.G.-G.; resources, D.J.; data curation, R.D.W. and C.G.-G. writing—original draft preparation, R.D.W., C.G.-G. and D.J.; writing—review and editing, D.J. and S.S.G.; visualization, R.D.W.; supervision, D.J. and C.G.-G.; project administration, D.J. All authors have read and agreed to the published version of the manuscript.

**Funding:** This research received no external funding.

**Institutional Review Board Statement:** Ethical review and approval were waived due to the nature of the study, that did not concern or expose any personal information.

**Informed Consent Statement:** Informed consent was obtained from all subjects involved in the study.

**Data Availability Statement:** Data is contained within the article and the Supplementary Material.

**Conflicts of Interest:** The authors declare no conflict of interest.

## Appendix A

Theoretical framework: The literature review uncovered a relation between the success of integrated, experimental, design-by-user, and participatory methods of learning in sustainable agriculture. There is a need for a progressive learning method, something that is in stark contrast to the often-used traditional classroom methods and top-down transfer of knowledge. In a response, Cooreman et al. (2018) developed a framework of peer learning between farmers in light of achieving sustainable agriculture [103]. This framework was chosen as it includes and confirms the important methods and factors described in the literature review. In addition to this, the framework goes deeper in-depth on the pedagogical effects that a training should aim to achieve. The core processes of this framework are engagement, communication initiation, and interactive knowledge creation. These three processes were carefully formed after extensive research had been performed in the subfields of adult learning, peer learning, and learning for sustainable development in agriculture by Cooreman et al. (2018). The engagement process is based on theories that formulated ownership, participation, trust, and informality as the key aspects. The process of communication is similarly built on theories and define sharing knowledge, formulating own values, and formulating questions as the key aspects. Finally, the process of interactive knowledge creation is built upon theories defining hands-on opportunities,

knowledge scaffolding, open discussion, and negotiating conflict (to arrive at a consensus) as its key aspects.

These three processes work interactively and create four learning outcomes: cognitive conflict, single-loop learning, double-loop learning, adoption and diffusion. Cognitive conflict occurs when new knowledge causes the learner to doubt his or her prior knowledge or to discover a certain lack of knowledge [103]. This phenomenon leads a learner to critically reflect upon their work and reframe their assumptions. Single-loop learning refers to acquiring factual knowledge and developing skills in order to manage problems on a daily basis. Building on this, double-loop learning explores the underlying values and assumptions, and requires critical reflection on the processes by which learning takes place [103]. This is a deeper form of learning in which the learner becomes aware of their own thinking and learning by asking critical questions such as: "Why is my farming system the way it is and should I change my farming system?".

Finally, the learning process comes to adoption and diffusion. Here, the farmer adopts the lessons learned and/or shares the acquired knowledge with his peers. This form of peer learning has consistently been the most common source of new information and ideas among farmers as they tend to be most influenced by proof of successful farming methods that is showed and explained by other farmers [103].

The three learning processes, and their learning outcomes, were used as a comparison/contrast to the data collected from desk research and the Delphi method, thus serving as the basis of this research.

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
