# Peer review of "Best Practices for Training in Sustainable Greenhouse Horticulture"

_sustainability, doi:10.3390/su15075816_

Round 1
Reviewer 1 Report
Dear Editor
Thank you for giving me the chance to tell you what I think about the article "Best practices for training in sustainable greenhouse horticulture." Please forgive me for taking so long to rate the article.
The manuscript addresses crucial issues regarding how to teach the concept of sustainable horticulture to all farming professionals in the province of Almeria in the south-east of Spain, one of the most important greenhouse areas for horticulture. How can the pedagogical methods of training in sustainability be improved in the greenhouse horticulture industry in Almeria is the focus of these studies. The paper provides new information regarding the challenges of teaching sustainable agriculture principles in greenhouse production.
However, more new references should be included in the introduction and used to compare their results with those of other researchers. In an effort to improve the sustainability of greenhouse horticulture in Almeria, the authors describe the most effective pedagogical practices and techniques for transferring technology and knowledge. The title and abstract accurately describe the manuscript's contents. The methods employed are contemporary, applicable, and sufficient to prove the tasks posed in the work. More comparisons with literature data are required, in my opinion. The manuscript is adequately illustrated and documented. The figures are of superior caliber.
This article should be published in the journal Sustainablity due to the significance of greenhouse production in Almeria, Spain, the negative effects of intensive production systems in addition to social issues, and the high priority of training farmers to achieve sustainable agriculture.
My Kind Regards
Author Response
Dear Reviewer,
Thank you very much for your critical evaluation of our manuscript. We gratefully followed up your suggestion to include new references from the recent literature into the introduction of the manuscript, and contrast these with our results. Subsequently, in the introduction section we added the following:
In line 100, we introduced a recent reference on the use of virtual education in sustainable horticulture.(Kee et al., 2022), which is now the new reference [30]
line 120: We introduced the following :The need for training in sustainable agriculture and precision agriculture technologies of farmers in Mediterranean greenhouses has also been identified in Greece (Kavga et al., 2001) as new reference [37].
The latter one was also cited in the discussion to contrast with our results (lines 872) : “A recent study in Greece suggests that elderly greenhouse farmers, with a low level of education, lack innovation culture and show a distrust in innovative trainings. In contrast to young graduates that have returned to agriculture, and which are open to training activities and innovation [37]”
Thank you again for your valuable suggestions.
Yours sincerely
Reviewer 2 Report
The authors put forward several principles to which agriculture has to adhere for it 866 to be sustainable. This paper is interesting and worth-of-interest because greenhouse horticulture is one of the main causes of waste production and energy utilization. The methodology is clear, and English is fine. I require some minor changes before definitively accepting it.
My comments are the following:
1. Avoid multiple references along the text, e.g. in lines 40 and 761. In addition, in line 696 there is a mistake in [49,49] while it should be [48,49].
2. Edit Table 4 and Table 5 as tables and not as images. In addition, all the tables are not formatted following the authors’ guidelines.
3. The paper pagination should be reviewed before final submission because there are some “white” lines along the text, i.e. lines 214-225, 271-281, 283-295, 297-312.
4. The quality of Figure A1 should be increased because it appears not clear for the reader.
5. The section “3. Results” should appear in the text with bold style and not normal style.
6. The sub-questions reported in lines 142-172 should be moved out from the Materials and Methods section and moved to the Introduction section, as always happen in scientific papers. In addition, I suggest creating a new section concerning the aim and scope of the paper.
7. As concern the structure, all the results obtained should be anticipated in the Materials and Methods section. In addition, the Discussion section should be summarized because it is long and difficult to follow.
8. I suggest the authors creating a bullet list to summarize the main results in the Conclusion section.
9. I did not find any limitations and future development of this research. I ask to add them into the revised paper.
Author Response
Dear Reviewer,
Thank you very much for your evaluation and your suggestions. Here we are answering (In talics) the questions and comments point-to-point:
- Avoid multiple references along the text, e.g. in lines 40 and 761. In addition, in line 696 there is a mistake in [49,49] while it should be [48,49].
This has now been corrected
- Edit Table 4 and Table 5 as tables and not as images. In addition, all the tables are not formatted following the authors’ guidelines.
All tables have now been formatted following the guidelines of the journal.
- The paper pagination should be reviewed before final submission because there are some “white” lines along the text, i.e. lines 214-225, 271-281, 283-295, 297-312.
This now been corrected
- The quality of Figure A1 should be increased because it appears not clear for the reader.
In the new version we have removed the Figure A1 because it is not original and is not essential for the comprehension of the section.
- The section “3. Results” should appear in the text with bold style and not normal style.
This has now been corrected.
- The sub-questions reported in lines 142-172 should be moved out from the Materials and Methods section and moved to the Introduction section, as always happen in scientific papers. In addition, I suggest creating a new section concerning the aim and scope of the paper.
This is an interesting suggestion. Actually, the questions asked in the paper are summarized at the end of the introduction. In the Materials and Methods section they are defined in detail.
- As concern the structure, all the results obtained should be anticipated in the Materials and Methods section. In addition, the Discussion section should be summarized because it is long and difficult to follow.
Thank you for the suggestion. The Discussion section follows the order of the results, which are referred to accordingly.
- I suggest the authors creating a bullet list to summarize the main results in the Conclusion section.
Has now been done.
- I did not find any limitations and future development of this research. I ask to add them into the revised paper.
Limitations have been written and introduced as the next to last paragraph in the Conclusion section.
Thank you again for your valuable suggestions,
Yours sincerely,
Reviewer 3 Report
Very good understanding is present for contemporary education of modern agrarians - students and farmers.
1. What is the main question addressed by the research?
Which is the best approach to give new information in the professional area of intensive glasshouse production.
2. Do you consider the topic original or relevant in the field? Does it
address a specific gap in the field?
The topic is original and is a good way to fill existing gaps. Info obtained in the bigger glasshouse area in Europe good be transferred all around EC.
3. What does it add to the subject area compared with other published
material?
It prove that in our world the classrooms are not the preferred place for obtaining studies. Many people know itq but now it is proven by a scientific survey.
4. What specific improvements should the authors consider regarding the
methodology? What further controls should be considered?
The work could be improved it new farmers included in the surveys. It is not easy because people do not like fill some questionnaires. It is my experience. The methodology could be enlarge in case of necessity.
5. Are the conclusions consistent with the evidence and arguments presented
and do they address the main question posed?
Conclusions are consistent with the evidence and arguments presented and do they address the main question posed
6. Are the references appropriate?
Yes – 101.
7. Please include any additional comments on the tables and figures.
For me data are well presented according different methodology. It is easy to understand.
8. Presented data nearly almost covers my own model for educational approach. It makes me happy! We can not stop classrooms presentations, but we must shrink them to 20-30% of time. But education will be more expensive.
Author Response
Dear Reviewer,
Thank you very much for your interesting comments. We are very satisfied to learn that you have similar experiences and opinions in this field.
Yours sincerely.